# Thoracic Spinal Sclerosing Epithelioid Fibrosarcoma Mimicking Schwannoma: Case Report and Literature Review

**DOI:** 10.3390/curroncol32110628

**Published:** 2025-11-07

**Authors:** Donato Creatura, Jad El Choueiri, Alberto Benato, Leonardo Anselmi, Ali Baram, Mario De Robertis, Carlo Brembilla, Federico Pessina, Maurizio Fornari, Gabriele Capo

**Affiliations:** 1Department of Biomedical Sciences, Humanitas University, Via Rita Levi Montalcini 4, Pieve Emanuele, 20090 Milan, Italy; 2Department of Neurosurgery, IRCSS Humanitas Research Hospital, Via Manzoni 56, Rozzano, 20089 Milan, Italy; 3Humanitas University, Via Rita Levi Montalcini 4, Pieve Emanuele, 20090 Milan, Italy; jad.elchoueiri@st.hunimed.eu; 4Northwestern Department of Neurological Surgery, 259 E Erie St 19th Floor, Chicago, IL 60611, USA

**Keywords:** sclerosing epithelioid fibrosarcoma, spine, spinal tumor, spinal cord tumor, MUC4, EWSR1, proton therapy

## Abstract

**Simple Summary:**

In this report, we describe the case of a young woman diagnosed with a thoracic foraminal lesion, initially suspected to be a common schwannoma. Following surgical resection, the lesion was identified as a rare malignant tumor, sclerosing epithelioid fibrosarcoma. The patient subsequently developed a recurrence, which required a second surgery and adjuvant proton therapy. By presenting this case alongside a review of previously published reports, our study underscores how this rare neoplasm can closely mimic more common spinal tumors, resulting in diagnostic delays or misclassification. We further highlight the need for a multidisciplinary management strategy and long-term surveillance. The insights gained from this work may assist spine surgeons in refining diagnostic accuracy and therapeutic decision-making for similarly rare spinal tumors.

**Abstract:**

Background/Objectives: Sclerosing epithelioid fibrosarcoma (SEF) is a rare soft tissue sarcoma with high rates of local recurrence and distant metastasis. Primary spinal involvement is exceedingly uncommon and often misdiagnosed due to radiological and histopathological resemblance to more frequent spinal tumors. The objective of this study is to present a rare case of thoracic spinal SEF and to contextualize it within the available literature. Methods: We describe the case of a 37-year-old woman presenting with progressive back pain and dysesthesia. MRI demonstrated a heterogeneously enhancing mass at the left T10–T11 neural foramen, initially interpreted as a common nerve sheath tumor. Gross total resection (GTR) was achieved, and histopathological analysis revealed a SEF. Clinical course, adjuvant therapies, and outcomes were evaluated, together with a review of previously reported spinal SEF cases. Results: Despite GTR followed by adjuvant chemotherapy, local recurrence occurred 18 months later. The patient underwent subtotal resection (STR) with adjuvant proton therapy. At 18-month follow-up after the second procedure, she remained neurologically stable and disease-free. The literature review confirmed the rarity of spinal SEF, its frequent misdiagnosis, and the absence of standardized therapeutic protocols. Conclusions: Spinal SEF is a rare malignancy that can mimic benign spinal tumors, delaying diagnosis. Its management relies on multidisciplinary assessment, individualized therapy, and long-term follow-up. This report increases awareness of spinal SEF and provides additional evidence to support clinical decision-making in rare spinal tumors.

## 1. Introduction

Spinal intradural-extramedullary and extradural lesions extending into the neural foramen represent a well-recognized diagnostic challenge. Among these, nerve sheath tumors—particularly schwannomas—constitute the most frequent consideration. They typically display the characteristic “dumbbell-shaped” configuration, with widening of the neural foramen and progressive compression of adjacent neural structures.

The differential diagnosis, however, is considerably broader and encompasses meningiomas with foraminal extension, metastatic disease, lymphomas, neurofibromas, germ cell tumors, primary sarcomas, and less common entities such as inflammatory or granulomatous processes or vascular tumors. Radiologically, these entities often exhibit overlapping features such as heterogeneous contrast enhancement or osseous remodeling, making accurate preoperative characterization inherently difficult.

Given that therapeutic strategies and prognosis vary substantially across this spectrum of pathologies, establishing an accurate diagnosis is paramount. The integration of clinical, radiological, and histopathological data therefore remains the cornerstone of management, while atypical or unexpected findings underscore the importance of a multidisciplinary approach involving neuroradiologists, spine surgeons, and pathologists.

Sclerosing epithelioid fibrosarcoma (SEF) is a very rare, aggressive soft tissue tumor, comprising only a small fraction of all sarcomas—fibrosarcoma itself accounts for just 1–2% of cases. SEF occurs most frequently in adults (average age 45–47 years) with near-equal gender distribution, though adolescent onset has been reported [1]. While typically localized to limb girdles, trunk, and extremities, SEF can arise virtually anywhere, including bone and spine. Wide surgical excision remains the mainstay of therapy whenever technically feasible. However, these tumors are prone to local recurrence and distant metastases, with reported rates as high as 40–86% [2]. Radiotherapy may be used for unresectable lesions or residual disease, but its efficacy is not definitively established. Conventional chemotherapy (e.g., doxorubicin/ifosfamide) generally shows limited benefit, and novel therapies remain investigational. Long-term follow-up is crucial due to the risk of late recurrence and metastasis [3,4,5].

In this report, we describe a rare case of thoracic spinal SEF with intra-, extradural, and foraminal extension, initially misdiagnosed as a schwannoma. We outline the multimodal treatment strategy adopted and provide a comprehensive review of the published cases of spinal SEF to refine current knowledge regarding its clinical presentation, diagnostic pitfalls, and management considerations. To date, only a handful of spinal SEF cases have been documented, predominantly as isolated case reports lacking standardized therapeutic protocols and long-term outcome data.

## 2. Case Report

A 37-year-old woman presented with a 6-month history of lower back pain radiating to both lower limbs, accompanied by progressively worsening dysesthesia. There was no history of trauma, neoplastic disease, or constitutional symptoms.

Neurological examination revealed isolated hypoesthesia in the left T10–T11 dermatomes, without objective motor deficits, gait disturbances, or sphincter dysfunction.

Magnetic resonance imaging (MRI) of the spine revealed a heterogeneously contrast-enhancing intra- and extracanalicular mass involving the left T10–T11 neural foramen. The lesion measured 21 × 8 × 21 mm and exhibited isointense signal on T1-weighted images and mildly hyperintense signal on T2-weighted images. The mass demonstrated both extradural and intradural components, compressing the spinal cord, and was initially considered a possible nerve sheath tumor given its location and partial extension into the neural foramen. (see Figure 1).

### 2.1. Surgical Intervention

In June 2022, the patient underwent a T10 laminectomy and partial left arthrectomy with GTR of the lesion under intraoperative neurophysiological monitoring (IONM). Intraoperatively, the lesion appeared predominantly extradural but was noted to infiltrate the dura and extend intradurally (see Figure 2). Dissection was challenging due to tumor infiltration of the dorsal nerve roots and its adherence to the pial surface of the spinal cord. GTR was achieved through sacrifice of the involved dorsal roots. The infiltrated dura was excised, and a patch of dural substitute (DuraGen^®^) was applied to the inner surface of the remaining dura mater (see Figure 2). The dura was then closed using 6-0 Prolene sutures and MRI-compatible ministaples, as previously described by Anselmi et al. [6]. An additional layer of TachoSil was applied extradurally to reinforce the dural closure.

### 2.2. Histopathological Findings

Histological analysis revealed features consistent with sclerosing epithelioid fibrosarcoma (grade 2, FNCLCC), characterized by nests and cords of epithelioid cells within a densely collagenous stroma, with focal necrosis and a moderate mitotic index. Immunohistochemistry was positive for MUC4, and the presence of an EWSR1 rearrangement was confirmed by FISH, supporting the diagnosis.

### 2.3. Adjuvant Therapy and Clinical Course

The postoperative course was uneventful, and the patient experienced significant improvement in preoperative symptoms. Immediate postoperative MRI demonstrated a GTR of the lesion, with no evidence of residual disease or complications. Adjuvant chemotherapy, consisting of three cycles of adriamycin and ifosfamide, was administered from August to September 2022 after multidisciplinary case discussion. Conventional radiotherapy and proton therapy were not considered due to the achieved GTR, absence of residual tumor, and close proximity to the spinal cord.

Radiological surveillance was performed at 2, 6, 12, and 18 months postoperatively with MRI. Follow-up MRI performed 18 months later revealed a recurrent lesion measuring 9 × 5 × 23 mm, in close contact with the spinal cord, associated with a T2 hyperintensity in the adjacent cord (see Figure 3).

A second surgery under IONM was performed in January 2024. Intraoperative findings confirmed a recurrent lesion adherent to the spinal cord. Subtotal resection (STR) was achieved to preserve neurological function. Histopathological findings confirmed recurrent SEF.

From February to March 2024, the patient underwent adjuvant proton therapy using the active scanning multifield optimization technique. Two clinical target volumes (CTVs) were irradiated, corresponding to low- and high-risk areas for disease recurrence, which received 50 Gy (relative biological effectiveness, RBE) in 25 fractions of 2 Gy (RBE) and 56 Gy (RBE) in 28 fractions of 2 Gy (RBE), respectively. The treatment was well tolerated, with no adverse effects or neurological deterioration.

### 2.4. Outcome and Follow-Up

At the 18-month follow-up after the second surgery (July 2025), the patient remained neurologically stable, with no motor deficits in the lower limbs, no gait disturbances, and no sphincter dysfunction. She was functionally independent and classified as grade I on the modified McCormick Scale. A residual hypoesthesia was noted in the left T10–T11 dermatomes. Follow-up imaging showed no evidence of disease recurrence.

## 3. Discussion

In this article, we report the case of a 37-year-old woman affected by a thoracic spinal lesion initially radiologically suspected to be a schwannoma, who underwent gross total resection. Histopathology unexpectedly revealed sclerosing epithelioid fibrosarcoma (SEF), and despite adjuvant chemotherapy, local recurrence developed within 18 months. A second surgery with subtotal resection was therefore performed, followed by adjuvant proton therapy. At the most recent follow-up, the patient remains neurologically intact and without evidence of disease. This case illustrates both the diagnostic pitfalls of spinal SEF, which may mimic benign lesions, and the therapeutic difficulties in balancing oncological radicality with neurological preservation.

In retrospect, it is conceivable that a more stepwise approach could have been pursued. Specifically, an early biopsy might have allowed a precise histopathological and molecular characterization prior to definitive surgery, thereby guiding a more oncologically radical resection strategy. In a hypothetical diagnostic flowchart, preoperative biopsy could indeed represent a valuable step to refine diagnosis and optimize surgical planning. However, in this case, a preoperative biopsy was not feasible due to the lesion’s intradural and intraspinal location, where the procedure would have entailed an unacceptably high risk of neurological injury.

Instead, initial GTR was performed based on a working diagnosis of schwannoma, and subsequent management, including chemotherapy and proton therapy, was tailored by a multidisciplinary team following unexpected histopathological findings. This underscores the importance of maintaining a high index of suspicion for rare malignant lesions even when imaging suggests a more common pathology and emphasizes individualized treatment planning in anatomically challenging spinal tumors.

### Rationale for Literature Review

Given the extreme rarity of spinal SEF and the lack of standardized therapeutic guidelines, we conducted a review of the literature. Particular attention was paid to the recurrent diagnostic challenges, as SEF often mimics more common spinal tumors both radiologically and histologically. By contextualizing our case within previously reported experiences, we aimed to highlight diagnostic pitfalls, outline current treatment strategies, and summarize long-term outcomes, thereby providing a broader framework for the management of this rare entity.

## 4. Literature Review

### 4.1. Methods

A targeted literature review was conducted on PubMed and Scopus (latest search June 2025) using the search query: (“sclerosing epithelioid fibrosarcoma”[Title/Abstract]) AND (spine[Title/Abstract] OR spinal[Title/Abstract] OR vertebra[Title/Abstract] OR vertebral[Title/Abstract] OR sacrum[Title/Abstract] OR sacral[Title/Abstract] OR thoracic[Title/Abstract] OR lumbar[Title/Abstract] OR cervical[Title/Abstract] OR paraspinal[Title/Abstract]), translated to each respective database. After removal of duplicates, the search yielded 13 results [7,8,9,10,11,12,13,14,15,16,17,18,19].

All case reports, case series, and relevant cohort studies were included, with no restriction on population. References of the included articles were manually checked for additional studies. Titles and abstracts were screened for relevance, and full texts were reviewed to confirm spinal involvement.

Data extracted from each report included patient demographics, tumor location, size, features, treatments, and outcomes. No formal meta-analysis was feasible; data were synthesized qualitatively.

### 4.2. Results

Our search identified 19 published cases of spinal SEF to date, drawn from individual case reports and small series. Table 1 summarizes the reported cases of spinal SEF, including patient characteristics, tumor location, treatment, and outcomes. Most identified cases were in adults. There was a slight female predominance, and tumors have been reported all throughout the spine.

In the following section, we provide a detailed synthesis of the available evidence, outlining the clinical, pathological, and therapeutic features of spinal sclerosing epithelioid fibrosarcoma (SEF) as reported in the literature.

SEF is a rare variant of fibrosarcoma that typically occurs in adults, characterized by cords and nests of monomorphic epithelioid cells embedded in a densely sclerotic extracellular matrix [15]. It was first described by Meis-Kindblom et al. in 1995 as a type of low-grade fibrosarcoma occurring primarily in the deep musculature with frequent association to the adjacent fascia or periosteum, and by the early 2000s, SEF was confirmed as a clinicopathologically distinct tumor with malignant potential and a high rate of recurrence and metastasis [17,20,21,22].

In 2002 and 2004, Abdulkader et al. and Chow et al., respectively, confirmed that SEF can occur as a primary bone tumor, with a clinical behavior probably similar to that of a soft tissue SEF, and a small number of cases have been documented over the past two decades on the pathology, with rare reporting of primary spinal SEF [7,17,23]. SEF is currently classified as a low- to intermediate-grade soft tissue sarcoma, characterized by high rates of local recurrence and distant metastasis. While SEF typically arises in the deep soft tissues of the extremities or trunk, primary involvement of the spine is exceedingly uncommon.

#### 4.2.1. Diagnostic Challenges

Because of its rarity and morphological overlap with other tumors, misdiagnosis of spinal SEF is not infrequent [15]. It may be confounded with metastatic carcinoma, osteosarcoma, hemangioendothelioma, Ewing sarcoma, or even benign lesions such as schwannomas due to similar radiological features [11,24]. Radiologically, spinal SEF often presents as a lytic, poorly marginated lesion, and histologically, its dense collagenous matrix may be misinterpreted as osteoid.

Imaging is usually consistent with a highly cellular tumor. On computed tomography, the tumor is typically a lytic, expansive lesion of bone mostly surrounded by sclerotic rims. Magnetic resonance usually shows a hypointense lesion in both T1- and T2-weighted sequences, often reported with peripheral contrast enhancement [18]. For instance, in a sacral SEF reported by Rocha et al., MRI demonstrated a lytic S1–S2 mass with extension into the spinal canal, hypointense on T1 and T2, and enhancement after gadolinium [9]. In the lumbar spine, SEF has been reported to cause collapse of the vertebral body and compressive deformity by Liu et al. [17]. Paraspinal soft tissue components can also occur [25]. While these findings strongly suggest a malignant process, they are not specific for SEF, contributing to the frequent misdiagnosis.

In our case, MRI led to an initial misdiagnosis of schwannoma. However, several features were atypical. The lesion showed heterogeneous contrast enhancement and involved the left T10–T11 neural foramen, which, unlike in most schwannomas, was not widened. Its signal characteristics were also uncharacteristic, as schwannomas are usually iso- or slightly hypointense relative to the spinal cord on T1-weighted images and hyperintense on T2-weighted images, sometimes with heterogeneous areas due to cystic degeneration. These findings, together with the ill-defined margins, should have raised suspicion for an alternative diagnosis, such as SEF, which typically appears heterogeneous, infiltrative, and poorly circumscribed, occasionally with bone involvement.

The diagnostic criteria of SEF remain mainly histological, consisting of cords, nests, and sheets of epithelioid cells embedded in sclerosed and hyalinized stroma. Righi et al. reported an interesting finding in a spinal case, showing a “hybrid” histology with classic SEF areas juxtaposed with regions displaying low-grade fibromyxoid sarcoma morphology [15]. Immunohistochemistry (IHC) plays a crucial role in diagnosis. Nearly all reported SEF cases demonstrate diffuse MUC4 positivity, a highly specific marker of SEF [16]. Cytoplasmic reactivity for vimentin has also been reported as a characteristic feature of SEF [15]. In addition, molecular techniques to identify FUS and EWSR1 rearrangements have evolved and are increasingly employed to confirm the diagnosis.

Integration of radiological and histological findings therefore remains crucial for accurate diagnosis and for distinguishing SEF from more common spinal entities, while recent advances in immunohistochemistry and molecular profiling have significantly improved diagnostic accuracy [15].

#### 4.2.2. Treatment and Outcomes

Despite its rarity, spinal SEF should be considered in the differential diagnosis of spinal tumors, as misdiagnosis or diagnostic delay remains a significant risk. From a management perspective, treating SEF in the spine should ideally entail complete surgical resection, although this may be limited by the proximity of the tumor to the spinal cord and nerve roots. Where feasible, a gross total resection (GTR) should be pursued, but the surgical approach must be individualized on a case-to-case basis.

Zhang et al. described an en bloc T1 spondylectomy with GTR for a cervicothoracic SEF case, although they noted that true wide margins are often not possible [11]. More recently, Liu et al. reported an L1 en bloc resection with negative margins, while Popli et al. achieved en bloc coccygectomy with clear margins [10,17]. All these patients remained disease-free at short- to mid-term follow-up.

In contrast, subtotal resection (STR) has been performed in anatomically constrained cases. Chow managed a sacral SEF with wide local excision after neoadjuvant radiotherapy, leaving residual disease [7]. Rocha et al. reported a 38-year-old with a sacral SEF treated with subtotal resection, followed by reoperation after three months and postoperative radiotherapy [9]. Mrimbaa described laminectomy and debulking at L3–L5 followed by chemotherapy due to extensive tumor involvement [19].

Radiotherapy (RT) has been employed both as neoadjuvant and adjuvant therapy. Lima planned neoadjuvant radiotherapy with a total dose of 50 Gy, followed by decompression and spinal fusion, and subsequently corpectomy, with no evidence of disease on MRI at 5 months [18]. The case reported by Chow also received a preoperative dose of 50 Gy [7]. Postoperative RT with a total dose of 60 Gy was administered by Rocha for residual disease [11]. Zhang also delivered adjuvant radiotherapy for local control after en bloc cervicothoracic spondylectomy, initiating treatment six months postoperatively to allow adequate spinal fusion, with a cumulative dose of 50 Gy [11]. In Righi’s spinal series, 4 out of 6 patients received adjuvant RT [15].

Chemotherapy has been inconsistently applied, and its role in SEF remains debated, as the tumor is generally considered poorly responsive to systemic treatment [26]. Santhosh et al. reported a case that did not respond to conventional sarcoma regimens, a finding consistent with SEF’s typically low proliferative activity and extensive sclerosis, which may further limit tumor vascularity [26]. Liu administered four cycles of doxorubicin following en bloc resection [17]. Mrimbaa reported adjuvant chemotherapy after subtotal resection [19]. Other spinal cases generally did not receive systemic therapy, unless initially misclassified as other sarcoma subtypes.

Recurrence and metastasis have been reported with variable patterns. SEF had long been described as a low-grade tumor. However, the 2013 WHO classification of tumors of soft tissue and bone included SEF among malignant fibroblastic/myofibroblastic tumors, reflecting, among other features, its propensity for local recurrence and late metastasis [9,27]. Chow’s patient developed pulmonary metastases at 5 years and died 8 years postoperatively [7]. In contrast, Popli, Zhang, and Liu reported no recurrence at follow-up intervals of 12–24 months [10,11,17]. Lima also reported no evidence of disease at 5 months [18]. Rocha and Mrimbaa did not provide long-term outcome data, although residual disease was present in both cases [9,19].

An outline of reported spinal SEF cases, including patient features, anatomical location, management, and results, is presented in Table 1.

### 4.3. Limitations

The main limitations of this review stem from the rarity of the condition and the reliance on heterogeneous case reports. A potential publication bias should be noted, as dramatic cases (with unusual features or poor outcomes) may be more often reported, while some long-term survivors or straightforward cases might go unpublished. Despite our comprehensive search strategy, the possibility remains that isolated or less visible reports may not have been captured. Nonetheless, by aggregating all available data, we provide the most comprehensive picture to date of SEF in the spine.

An international registry or further prospective multi-center studies would help deepen understanding and help set guidelines for the management of such tumors, with a focus on molecular-targeted treatments.

## 5. Conclusions

Sclerosing epithelioid fibrosarcoma (SEF) of the spine is a rare and diagnostically challenging entity, often mimicking more common spinal tumors both radiologically and histologically. Its infiltrative nature, propensity for local recurrence, and potential for distant metastasis underscore the importance of early recognition and a multidisciplinary approach to management.

Surgical resection remains the cornerstone of treatment, although gross total resection is frequently limited by anatomical constraints. Adjuvant therapies, including radiation and chemotherapy, may offer additional local control, though their efficacy remains to be fully elucidated.

Given the paucity of reported cases and lack of standardized treatment protocols, collaborative efforts and prospective multicenter registries are essential to better characterize the clinical behavior of spinal SEF and guide evidence-based therapeutic strategies.

## Figures and Tables

**Figure 1 curroncol-32-00628-f001:**
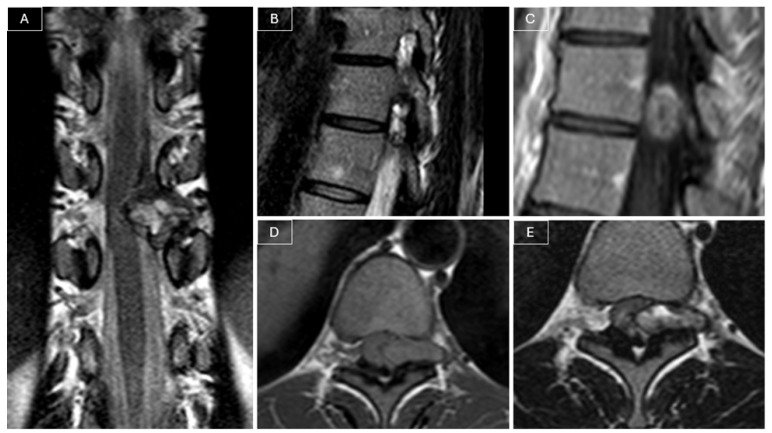
Preoperative MRI. (**A**) Coronal T2-weighted images demonstrated a heterogeneously hyperintense lesion compressing the spinal cord. The lesion appeared to have both intradural and extradural components and extended into the left T10–T11 neural foramen. (**B**) Sagittal T2-weighted images confirmed involvement of the left T10–T11 foramen by the tumor. (**C**) Sagittal post-contrast T1-weighted images showed the heterogeneous contrast enhancement of the lesion. The lesion appeared isointense on T1-weighted images (**D**) and slightly hyperintense on T2-weighted images (**E**).

**Figure 2 curroncol-32-00628-f002:**
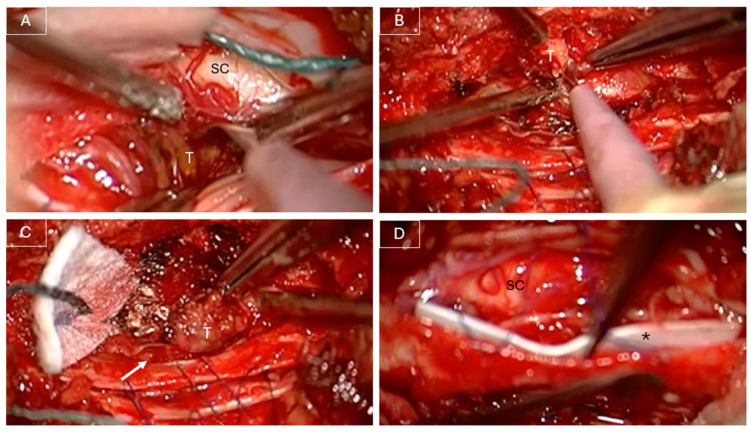
Intraoperative images. (**A**) View from the left, removing the intradural portion of the tumor (T); dissection from the lateral pial plane of the spinal cord (SC). (**B**,**C**) View from the right side, removing the tumor’s portion in the left T10–T11 foramen. The portion of the tumor in the left foramen was removed while holding the microscope on the right side of the spinal cord, without the need to enlarge the left partial arthrectomy. Note the dural gap (white arrow) in the lateral aspect of the dural sac after the infiltrated dura was excised. (**D**) DuraGen^®^ patch applied to the inner surface of the remaining dura mater. T = tumor. SC = spinal cord. White arrow = dural gap. * = patch of dural substitute.

**Figure 3 curroncol-32-00628-f003:**
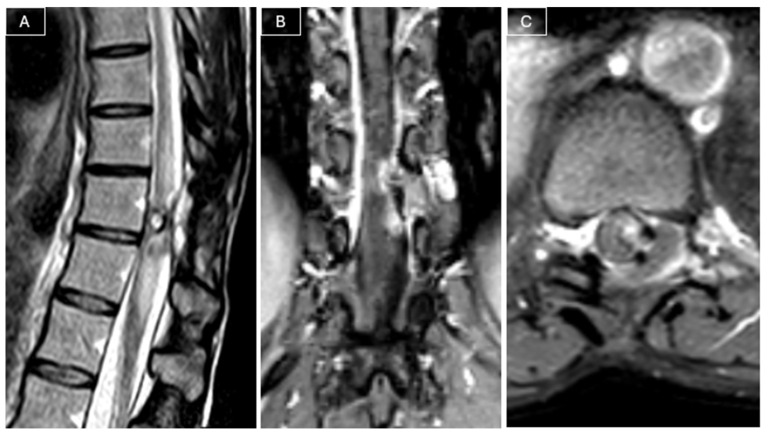
MRI at 18-month follow-up revealed tumor recurrence. (**A**) Sagittal T2-weighted images showed a T2 hyperintensity in the adjacent cord. (**B**,**C**) Post-contrast T1-weighted images showed a nodular contrast-enhancing lesion in contact with the spinal cord, suspicious for tumor recurrence.

**Table 1 curroncol-32-00628-t001:** Table summarizing previous studies on spinal SEF.

Author (Year)	Number of Patients	Biological Sex/Age	Spinal SEF Location	Treatment	Outcome/Follow-Up
Chow et al., 2004 [7]	1	F, 48	Sacral	Neoadjuvant RT, Surgery	Metastatic recurrence, survival 8 years
Wojcik et al., 2014 [8]	1 (of 8 bone SEF)	M, 66	Cervical (C6)	N.R.	N.R.
Rocha et al., 2017 [9]	1	F, 38	Sacral	STR, reoperation after 3 months, Adjuvant RT	No evidence of tumor regrowth at 12 months
Popli et al., 2018 [10]	1	F, 77	Coccyx	Surgery	No symptoms 2 years post-op
Zhang et al., 2018 [11]	1	M, 64	Cervicothoracic junction (C5-T1)	En bloc spondylectomy and GTR, Adjuvant RT	Temporary symptomatology, no evidence of residual tumor
Tsuda et al., 2020 [12]	1 (of 7 bone SEF)	M, 47	Thoracic	N.R.	N.R.
Porteus et al., 2020 [13]	1 (of 8 SEFs)	F, 45	Sacrum	N.R.	Alive at 10 months
Kosemehmetoglu et al., 2021 [14]	2 (of 9 bone SEF)	F, 22 F, 51	Cervicothoracic (C6-T1) Lumbosacral (L3-S1)	Surgery, RT, Chemotherapy RT, Chemotherapy	Alive with disease (recent) Alive with disease (1 year)
Righi et al., 2021 [15]	6	3M 3F Median Age: 41	1 Cervical 2 Thoracic 3 Lumbar	Surgery Chemotherapy (4) RT (3)	Local recurrence (2) Dead of disease (3) Dead of complications (1) Alive with disease (1)
Wei et al., 2022 [16]	1	F, 28	Thoracic	Surgery	No signs of recurrence or metastasis at 12 months
Liu et al., 2023 [17]	1	F, 61	Lumbar	Surgery	No signs of recurrence at 12 months
Lima et al., 2023 [18]	1	F, 46	Lumbar (L5)	Neoadjuvant RT, Surgery	Right sciatica at 4 months, No sign of disease persistence at 5 months
Mrimba et al., 2024 [19]	1	M, 35	Lumbar (L3-L5)	Surgery, Adjuvant chemotherapy	Regression with minimal residue at 6 months follow-up, Neurologically intact 1-year post-op

F = Female; M = Male; SEF = Sclerosing Epithelioid Fibrosarcoma; N.R. = Not Reported; STR = Subtotal Resection; GTR = Gross Total Resection; RT = Radiotherapy.

## Data Availability

The original contributions presented in this study are included in the article material. Further inquiries can be directed to the corresponding author.

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
