# Peer review of "Thoracic Spinal Sclerosing Epithelioid Fibrosarcoma Mimicking Schwannoma: Case Report and Literature Review"

_curroncol, 2025, doi:10.3390/curroncol32110628_

Round 1
Reviewer 1 Report
Comments and Suggestions for Authors
Interesting and well-documented case. However, the MRI appearance was not typical for a schwannoma — it was heterogeneous, partly cystic, and lacked clear foraminal enlargement. Please rephrase the claim that imaging was “suggestive of schwannoma.” Also note that biopsy was not realistically feasible due to the intraspinal location. The discussion should focus less on novelty and more on management rationale and multidisciplinary approach.
Author Response
"Interesting and well-documented case. However, the MRI appearance was not typical for a schwannoma — it was heterogeneous, partly cystic, and lacked clear foraminal enlargement. Please rephrase the claim that imaging was “suggestive of schwannoma.” Also note that biopsy was not realistically feasible due to the intraspinal location. The discussion should focus less on novelty and more on management rationale and multidisciplinary approach."
Comment: The MRI appearance was not typical for a schwannoma — it was heterogeneous, partly cystic, and lacked clear foraminal enlargement. Please rephrase the claim that imaging was “suggestive of schwannoma.”
Response:
We agree. The wording has been revised throughout the manuscript to better reflect the atypical imaging features. The sentence in the Abstract and Case Report sections now reads:
“MRI demonstrated a heterogeneously enhancing mass at the left T10–T11 neural foramen, initially interpreted as a common nerve sheath tumor.”
“The mass demonstrated both extradural and intradural components, compressing the spinal cord, and was initially considered a possible nerve sheath tumor given its location and partial extension into the neural foramen”
This phrasing acknowledges diagnostic uncertainty and avoids overstatement.
Comment: Biopsy was not realistically feasible due to the intraspinal location.
Response:
We have added a clarifying sentence in the Discussion:
“In a hypothetical diagnostic flowchart, preoperative biopsy could indeed represent a valuable step to refine diagnosis and optimize surgical planning. However, in this case, a preoperative biopsy was not feasible due to the lesion’s intradural and intraspinal location, where the procedure would have entailed an unacceptably high risk of neurological injury”
Comment: The discussion should focus less on novelty and more on management rationale and multidisciplinary approach.
Response:
We have substantially revised the Discussion to emphasize the management rationale and multidisciplinary coordination among neurosurgery, oncology, and radiotherapy teams. Statements highlighting “novelty” have been reduced, and the focus now rests on clinical reasoning and interdisciplinary strategy.
Reviewer 2 Report
Comments and Suggestions for Authors
Dear Editor, dear authors
thank you very much for giving me the opportunity to review the case report “Thoracic Spinal Sclerosing Epithelioid Fibrosarcoma Mimicking Schwannoma: Case Report and Literature Review” submitted to the journal “Current Oncology”.
This is an interesting review presenting the special case of a thoracic fibrosarcoma mimicking a schwannoma.
However, this review has several limitations that should be addressed prior publication.
Major concerns:
- Abstract: Conclusion should be shortened
- Introduction: Authors should describe entity, prevalence and therapy of fibrosarcoma in more detail to give the readers mor background informations.
- Figure 1: Authors should ad saggital T1 with post-contrast
- Figure 2: (B) does not add any information to the reader and might be removed. Figure should be described in more detail. It is hard for the reader to find the tumor and its surrounding anatomy.
- Figure with the histopathological slices should be added.
- Please describe the chemotherapy protocol in more detail. Postoperative was not performed after the first surgery, why? Should be stated within the paragraph “Adjuvant therapy and clinical course. Which time interval was assessed for MRI controls?
- Review of the literature:
- Methods should be moved within the section methods, results of the review should be moved to the results of the article
- Discussion: Radiological characteristics of schwannoma should be added within the section diagnostic challenges and compared with the characteristics of SEF
- Conclusion should be shortened; Radiotherapy and adjuvant therapy may be presented in conclusion as an adjuvant therapy that should be performed.
Minor concerns:
Flow chart of the literature review should be added
Author Response
"Dear Editor, dear authors
thank you very much for giving me the opportunity to review the case report “Thoracic Spinal Sclerosing Epithelioid Fibrosarcoma Mimicking Schwannoma: Case Report and Literature Review” submitted to the journal “Current Oncology”. This is an interesting review presenting the special case of a thoracic fibrosarcoma mimicking a schwannoma. However, this review has several limitations that should be addressed prior publication.
Major concerns:
- Abstract: Conclusion should be shortened
- Introduction: Authors should describe entity, prevalence and therapy of fibrosarcoma in more detail to give the readers mor background informations.
- Figure 1: Authors should ad saggital T1 with post-contrast
- Figure 2: (B) does not add any information to the reader and might be removed. Figure should be described in more detail. It is hard for the reader to find the tumor and its surrounding anatomy.
- Figure with the histopathological slices should be added.
- Please describe the chemotherapy protocol in more detail. Postoperative was not performed after the first surgery, why? Should be stated within the paragraph “Adjuvant therapy and clinical course. Which time interval was assessed for MRI controls?
Review of the literature:
- Methods should be moved within the section methods, results of the review should be moved to the results of the article
- Discussion: Radiological characteristics of schwannoma should be added within the section diagnostic challenges and compared with the characteristics of SEF
- Conclusion should be shortened; Radiotherapy and adjuvant therapy may be presented in conclusion as an adjuvant therapy that should be performed.
Minor concerns: Flow chart of the literature review should be added."
Major Concerns
Comment: Abstract: Conclusion should be shortened.
Response:
The Abstract conclusion has been shortened to focus concisely on the main message:
“Spinal SEF is a rare malignancy that can mimic benign spinal tumors, delaying diagnosis. Its management relies on multidisciplinary assessment, individualized therapy, and long-term follow-up. This report increases awareness of spinal SEF and provides additional evidence to sup-port clinical decision-making in rare spinal tumors.”
Comment: Introduction: Describe entity, prevalence, and therapy of fibrosarcoma in more detail.
Response: We have added this paragraph to the introduction:
“Sclerosing epithelioid fibrosarcoma (SEF) is a very rare, aggressive soft tissue tumor, comprising only a small fraction of all sarcomas—fibrosarcoma itself accounts for just 1–2% of cases. SEF occurs most frequently in adults (average age 45–47 years) with near-equal gender distribution, though adolescent onset has been reported. While typical-ly localized to limb girdles, trunk, and extremities, SEF can arise virtually anywhere, in-cluding bone and the spine. Wide surgical excision remains the mainstay of therapy whenever technically feasible. However, these tumors are prone to local recurrence and distant metastases, with reported rates as high as 40–86%. Radiotherapy may be used for unresectable lesions or residual disease, but its efficacy is not definitively established. Conventional chemotherapy (e.g., doxorubicin/ifosfamide) generally shows limited bene-fit, and novel therapies remain investigational. Long-term follow-up is crucial due to the risk of late recurrence and metastasis”
Comment: Figure 1: Add sagittal T1 with post-contrast.
Response:
Thank you for your suggestion. We would like to kindly point out that a sagittal T1-weighted post-contrast image is already included as Figure 1C.
Comment: Figure 2: (B) does not add information; may be removed. Describe figure in more detail.
Response:
Thank you for your suggestion. We would like to kindly point out that Figure 2B was included to better illustrate how the portion of the tumor in the left foramen was removed while holding the microscope on the right side of the spinal cord, without the need to enlarge the left partial arthrectomy. The figure legend has been updated.
Comment: Add histopathological figure.
Response:
Unfortunately, we do not have availability of histopathological images for this case. We acknowledge the importance of including such figures for diagnostic clarity and regret this limitation. We have thoroughly described the histopathological findings in the text to compensate for this absence.
Comment: Describe chemotherapy protocol in more detail. Postoperative chemotherapy was not performed after the first surgery—why? Clarify MRI follow-up interval.
Response:
Thank you for your comment and suggestions. We had not previously mentioned the immediate postoperative MRI, which demonstrated a gross total resection (GTR), so we have now included this information. The adjuvant chemotherapy was administered after the first surgery, and the specific protocol used has been clarified; it was decided through a multidisciplinary board where radiotherapy or proton therapy was excluded due to the achieved GTR, absence of residual tumor, and proximity to the spinal cord. Radiological surveillance was performed at 2, 6, 12, and 18 months with MRI.
Comment: Review methods should be moved to Methods; results to Results.
Response:
Thank you for your comment.
However, we have created subsections (Methods and Results) within the Literature Review section, in order to avoid adding further sections to the manuscript.
Comment: Discussion: Compare radiological characteristics of schwannoma and SEF.
Response:
A dedicated paragraph has been added under 4.2.1 Diagnostic challenges, comparing MRI features of schwannoma (typically homogeneous, well-circumscribed, with foraminal enlargement) versus SEF (heterogeneous, infiltrative, less circumscribed, possible bone destruction).
Comment: Conclusion should be shortened; highlight adjuvant therapy.
Response:
The Conclusion has been condensed and explicitly mentions the potential role of adjuvant radiotherapy and chemotherapy in multimodal management.
Minor Concern: Add a flow chart for the literature review.
Response:
Thank you for your suggestion. However, as this was not a systematic literature review, we believe that including a flow diagram would not be appropriate in this context.
Reviewer 3 Report
Comments and Suggestions for Authors
This is a high-quality case report that meticulously documents a rare case of thoracic spinal sclerosing epithelioid fibrosarcoma (SEF) mimicking a schwannoma. It covers the clinical presentation, imaging features, surgical details, pathological diagnosis, adjuvant therapy, and follow-up outcomes. The discussion is comprehensive, incorporating a literature review that highlights diagnostic challenges, the importance of multidisciplinary management, and the need for long-term monitoring. This report provides valuable clinical insights for spinal surgeons and oncologists, particularly in handling similar rare spinal tumors.
Weaknesses:
- Insufficient Preoperative Imaging Data: While the article includes a description and illustration of preoperative MRI (Figure 1), it lacks additional details such as multi-sequence imaging or supplementary CT scans, which could better demonstrate bone destruction or vascular features. It is recommended to add more imaging evidence or explain why additional examinations were not performed.
- Inadequate Differentiation from Schwannoma: Although the article notes that SEF mimics schwannoma, the discussion on morphological differences (imaging and pathology) is not sufficiently in-depth. For example, further comparison of T1/T2 signal intensities, enhancement patterns, or molecular markers could strengthen diagnostic guidance.
- No Frozen Section During First Surgery: The article describes gross total resection (GTR) in the initial surgery but does not mention intraoperative frozen biopsy. This could lead to diagnostic delays, especially when a suspected benign tumor turns out to be malignant. It is suggested to include a retrospective analysis in the discussion, explaining the reasons for not performing frozen sections and discussing their potential benefits (e.g., guiding more aggressive resection).
- Other Minor Issues: The literature review is comprehensive but includes some older references; the follow-up period (18 months) is relatively short, and authors could suggest updating long-term outcomes in the future.
Author Response
"This is a high-quality case report that meticulously documents a rare case of thoracic spinal sclerosing epithelioid fibrosarcoma (SEF) mimicking a schwannoma. It covers the clinical presentation, imaging features, surgical details, pathological diagnosis, adjuvant therapy, and follow-up outcomes. The discussion is comprehensive, incorporating a literature review that highlights diagnostic challenges, the importance of multidisciplinary management, and the need for long-term monitoring. This report provides valuable clinical insights for spinal surgeons and oncologists, particularly in handling similar rare spinal tumors.
Weaknesses:
- Insufficient Preoperative Imaging Data: While the article includes a description and illustration of preoperative MRI (Figure 1), it lacks additional details such as multi-sequence imaging or supplementary CT scans, which could better demonstrate bone destruction or vascular features. It is recommended to add more imaging evidence or explain why additional examinations were not performed.
- Inadequate Differentiation from Schwannoma: Although the article notes that SEF mimics schwannoma, the discussion on morphological differences (imaging and pathology) is not sufficiently in-depth. For example, further comparison of T1/T2 signal intensities, enhancement patterns, or molecular markers could strengthen diagnostic guidance.
- No Frozen Section During First Surgery: The article describes gross total resection (GTR) in the initial surgery but does not mention intraoperative frozen biopsy. This could lead to diagnostic delays, especially when a suspected benign tumor turns out to be malignant. It is suggested to include a retrospective analysis in the discussion, explaining the reasons for not performing frozen sections and discussing their potential benefits (e.g., guiding more aggressive resection).
- Other Minor Issues: The literature review is comprehensive but includes some older references; the follow-up period (18 months) is relatively short, and authors could suggest updating long-term outcomes in the future."
Comment: Add more preoperative imaging details or explain why additional scans were not performed.
Response:
CT was not performed preoperatively, as MRI findings were deemed sufficient for surgical planning and there was no suspicion of bone destruction or vascular involvement.
Comment: Enhance discussion on differentiation from schwannoma (imaging and pathology).
Response:
The Discussion now includes a more detailed differential comparison between SEF and schwannoma, focusing on MRI signal intensities and enhancement patterns.
Comment: Explain why no frozen section was done during first surgery and discuss its potential role.
Response:
Intraoperative frozen section was not performed. Retrospectively, its use could have facilitated earlier recognition of malignancy.
Comment: Update older references and suggest long-term follow-up.
Response:
Thank you for your valuable comment. We appreciate the suggestion and agree that longer follow-up and updated long-term outcome data would further strengthen the study. Several recent references have been added.
Round 2
Reviewer 2 Report
Comments and Suggestions for Authors
Authors adopted most of the suggestions.